# Optimization of the Polarization Profile of Conical-Shaped Shells Piezoelectric Sensors

**DOI:** 10.3390/s23010442

**Published:** 2022-12-31

**Authors:** Sergio Horta Muñoz, David Ruiz

**Affiliations:** 1Instituto de Investigación Aplicada a la Industria Aeronáutica, Escuela de Ingeniería Industrial y Aeroespacial de Toledo, Universidad de Castilla-La Mancha, Campus Fábrica de Armas, Av. Carlos III, 45004 Toledo, Spain; 2OMEVA Research Group, Escuela de Ingeniería Industrial y Aeroespacial de Toledo, Universidad de Castilla-La Mancha, Campus Fábrica de Armas, Av. Carlos III, 45004 Toledo, Spain

**Keywords:** topology optimization, piezoelectric sensors, conical shell, finite element method

## Abstract

Conical shell structures are frequently submitted to severe static and dynamic mechanical loads that can result in situations that affect the service of the systems that are part of, or even cause catastrophic failures. For this reason, a common solution is to design an active deformation control system, usually using piezoelectric patches strategically distributed along the surface of the shell structure. Moreover, these elements may be part of an energy recovery system. This paper details the methodology to topologically optimize the placement of piezoelectric elements through a characteristic function, analysing static and free vibration loading cases by means of the finite element method. Then, the optimal arrangement of the electrode with different polarization profiles is distributed throughout the entire structure. The nature of the loading cases studied corresponds to a general situation where static loads and dynamics vibration are considered. The objective function of the problem only depends linearly on the displacement fields, and therefore, the optimal electrode profile can be obtained for any combination of loads. As a consequence, this technique allows for maximising the electric charge obtained, which results in a greater capacity for monitoring, actuation and/or energy harvesting.

## 1. Introduction

Conical-shaped shells appear frequently in industrial applications, mainly in elements which have a relevant mechanical-structural function in sectors such as military, civil, nuclear and aerospace engineering. Specifically, highlighting aerospace applications, it is common to find truncated cone-shaped elements forming part of propulsion systems (turbine engines, injectors, nozzles) as well as fuselage structural elements (rocket fairing, droop nose in supersonic aircraft) [1,2]. These elements are generally subjected to large thermomechanical loadings, both static and dynamic (i.e., pressure loads distributed on their surface, vibrations, inertial forces, etc.), which cause severe deformations and can lead to aeroelastic effects and, subsequently, loss of control in flight and catastrophic failures. Therefore, it is necessary to carry out regulation of the vibration of these elements, in applications in which the mechanical properties can be found fixed by other material requirements, for instance, weight reduction and extreme environmental conditions. Then, it is necessary to carry out monitoring and, sometimes, acting on the movement to dampen oscillations. Piezoelectric sensors and actuators are a frequent solution in these practices [3,4,5].

Not only is the application of piezoelectrics interesting for mechanical performance, but also for energy harvesting, which is of great interest in a current context in which the aerospace sector seeks green technologies free of fossil fuels. In this sense, different authors [6,7] have sought the use of piezoelectric patches to obtain an energy supply that can feed low-power devices. Li et al. [7] performed an optimization of the size of the patches placed on conical shells. Firstly, the position where a mode reaches a larger strain was found, which results in higher modal voltage generated, and later the rectangular patch size that maximizes the generated voltage was determined. The theoretical development of the expressions is solved by applying the Rayleigh-Ritz numerical method. Aimmanee and Phongsitthisak [6] also carried out a theoretical study on the energy harvesting capacity of a conical shell, in this particular case on a Belleville’s spring, as this is an element continuously subjected to alternating compressive loads. The analytical model, also solved by the Rayleight-Ritz method, is contrasted with commercial software based on the finite element method (FEM), using a piezoelectric layer that covers the entire analysed structure.

From the mechanical point of view, most of the articles in this regard [1,4,8,9] are based on the study of the free vibration of conical shells, to act on the natural oscillation modes under different displacement constraints. However, it should be noted that Li et al. [10] analytically studied the forced vibrations of conical shells, obtaining the equations of motion under different external dynamic loads. Continuing this line, other works focus on using piezoelectric actuators to carry out active control of vibrations. Li et al. [3] developed a genetic algorithm for placing a determined number of piezoelectric patches that mitigate forced vibration deformations, emphasizing the linear quadratic regulator (LQR) scheme. However, the scope and methodology used for the optimization of the location of patches is not detailed in depth. A work that brings this subject to the experimental field is carried out by Sun et al. [11], where two pairs of piezoelectric ceramic patches are used as actuators/exciters and sensors. Numerical simulations of the natural vibrations of the structure are experimentally corroborated, and later an LQR is successfully applied to reduce the vibrations of a conical-shaped shell structure. However, the positioning of actuators and patches is determined arbitrarily, being located in areas where the amplitudes of the oscillations are higher.

Jamshidi and Jafari develop several works [5,8,9,12] on the subject of using piezoelectric actuators and sensors in conical shells for vibration control. In particular, [8,9] focused on designing the distribution of these piezoelectric elements through analytical relations based on the Kirchhoff–Love thin plate theory. Different patch distributions are applied in an ordered manner in longitudinal, circumferential or diagonal directions. However, the authors point out that these kinds of periodic distribution can reduce effectiveness, so they resort to the solution of having odd numbers of patches. In [13], the fluid loading effect in piezoelectric cylindrical shells for acoustic problems is modelled. Elhami et al. [14] study the wave propagation in air-filled double-walled cylindrical shells, including the piezoelectric effect (direct and inverse) to use piezoelectric patches as sensors and actuators, respectively.

Regarding the conical structures analysed in previous works, it should be noted that the reproduced boundary conditions vary within a series of typical cases, which can reproduce different real situations. Specifically, works have been found in the referenced bibliography that study cases of clamped-free conditions [3,11], but other authors [8,9] indicate that simply-supported conditions at both ends are more interesting for the industrial field.

In general, the reduction of the size of the sensor increases the range of applications, with the counterpart also reducing the signal produced by the device. The optimization of the electric load acquires special relevance then. Lee and Moon [15] carried out a pioneer work where piezoelectric transducers are applied. In this work, the distribution of patches is analytically studied for one-dimensional modal sensors. Donoso et al. [16,17] extended this work to modal filters, maximizing the electric charge through a linear optimization problem where the electrode profile is optimized. This idea, including the optimization of the host structure, is carried out in [18,19,20] for static and modal cases, respectively. This article aims to extend all these previous works to conical-shaped sensors. The analysis is stated as a topology optimization problem, where the electrode profile is not restricted to simply shaped patches, as in previously mentioned studies. The problem proposed lacks constraints over the shape of the electrodes obtained, as this is out of the scope of the paper. If the reader is interested in taking into consideration manufacturing constraints, the method proposed in [21] can be implemented. In this paper, the authors present an eigenvalue problem based on the graph theory to ensure the connectivity of each of the phases.

This work is divided into sections as follows. Section 2 introduces the continuous formulation and the finite element model of the problem to be studied. The discretized optimization problem is described in Section 3. Section 4 is focused on analysing the optimal designs obtained. To conclude, the conclusion of this work is presented in Section 5.

## 2. Generation of Electric Charge

### 2.1. Piezoelectric Sensor

Piezoelectricity is the ability to generate an electric charge *q* in response to a mechanical strain. Considering the piezoelectric isotropy ($e_{31}=e_{32}$) of the sensor, and the negligible effect of the piezoelectric layer and the electrodes in the mechanical response of the structure, this value *q* is computed following the Equation (Equation 1) [15].
(1)$$\begin{array}{l} q=\int_{\Omega }\chi \left(x_{1},x_{2}\right)\left[e_{31}\varepsilon_{11}+e_{32}\varepsilon_{22}\right]d\Omega =\\ \;\;\;\;\;\;\;\;\;\;\;\;\;\;\;\;\;\;\;\;\;\;\;\;\;\;\;\;\;\;\;\;\;\;\;\;\;\;\;\;\;\;\;\;=e_{31}\int_{\Omega }\chi \left(x_{1},x_{2}\right)\left[\frac{\partial u_{1}}{\partial x_{1}}+\frac{\partial u_{2}}{\partial x_{2}}+x_{3}\left(\frac{\partial^{2}u_{3}}{\partial {x_{1}}^{2}}+\frac{\partial^{2}u_{3}}{\partial {x_{2}}^{2}}\right)\right]d\Omega ,\; \end{array}$$
where Ω is the design domain, χ∈{−1,0,1} is a tri-level function that models the electrode polarity, $e_{31}$ is the piezoelectric constant, $\varepsilon_{11}$ and $\varepsilon_{22}$ are the in-plane normal strains, $\left(u_{1},u_{2}\right)$ are the translational in-plane displacements and $u_{3}$ the displacement normal to the shell mid-surface.

The displacement and rotation fields must be computed by solving the equilibrium equation, which is different for each case study.

### 2.2. Static Case

For the first case study, the following time-independent problem is solved:$$\left\{\begin{array}{c} \begin{array}{cccc} -\nabla \cdot \left(E_{s}:\varepsilon \right) & = & f_{v}, & \mathrm{in}\;\Omega \\ \left(E_{s}:\varepsilon \right)\cdot n & = & f_{s}, & \mathrm{in}\;\Gamma_{f}\\ u_{1},u_{2},u_{3} & = & 0, & \mathrm{in}\;\Gamma_{c}\; \end{array} \end{array}\right.,$$
where $\left(u_{1},u_{2},u_{3}\right)$ are the displacements in each axis, $E_{s}$ and ε are the stiffness and the infinitesimal strain tensors, and $\left(f_{s},f_{v}\right)$ the surface and volumetric forces, respectively. Null Dirichlet conditions are applied over the boundary $\Gamma_{c}$.

### 2.3. Vibration Modes

In this second study case, the vibration modes of the structure need to be computed through the next equation:$$\begin{array}{cccc} \nabla \cdot \left(E_{s}:\tilde{\varepsilon }\right) & = & \rho \;\ddot{u}, & \mathrm{in}\;\Omega ,\;\; \end{array}$$
subject to the boundary conditions:$$\begin{array}{cccc} {\bar{u}}_{1},{\bar{u}}_{2},{\bar{u}}_{3} & = & 0, & \mathrm{in}\;\Gamma_{c},\; \end{array}$$
and:$$\left\{\begin{array}{c} \begin{array}{ccc} {\bar{u}}_{1} & = & \sum_{i=1}^{\infty }u_{1i}\left(x_{1},x_{2}\right)\eta_{u_{1}i}\left(t\right)\;\\ {\bar{u}}_{2} & = & \sum_{i=1}^{\infty }u_{2i}\left(x_{1},x_{2}\right)\eta_{u_{2}i}\left(t\right)\;\\ {\bar{u}}_{3} & = & \sum_{i=1}^{\infty }u_{3i}\left(x_{1},x_{2}\right)\eta_{u_{3}i}\left(t\right)\; \end{array}, \end{array}\right.$$
where $\left({\bar{u}}_{1},{\bar{u}}_{2},{\bar{u}}_{3}\right)$ represent the vibration modes. Using modal decomposition, they are divided into the mode shapes $\left(u_{1i},u_{2i},u_{3i}\right)$ and the modal coordinates $\left(\eta_{u_{1}i},\eta_{u_{2}i},\eta_{u_{3}i}\right)$, where the spatial part is used to compute the electric charge in Equation (Equation 1). The subscript *i* indicates the *i*-th vibration mode.

### 2.4. Finite Element Model

Thin-shell formulation is developed based on Kirchhoff-Love plate theory for a bi-dimensional element consisting of 4 nodes with 6 degrees of freedom (DOF), 3 displacements $u_{1}$, $u_{2}$ and $u_{3}$, and 3 rotations $\phi_{1}$, $\phi_{2}$ and $\phi_{3}$ [22,23], described in terms of an element local coordinate system $\left(x_{1},x_{2},x_{3}\right)$. Rotations $\phi_{1}$ and $\phi_{2}$ could be related to $u_{3}$, the displacement in the normal direction, by means of their partial derivatives:(2)$$\left[\begin{array}{c} \phi_{1}\\ \phi_{2} \end{array}\right]=\left[\begin{array}{cc} 0 & 1\\ -1 & 0 \end{array}\right]\left[\begin{array}{c} \frac{\partial u_{3}}{\partial x_{1}}\;\\ \frac{\partial u_{3}}{\partial x_{2}} \end{array}\right].$$

The interpolation of in-plane displacements, associated with membrane behaviour, is described in Equation (Equation 3).
(3)$$\left[\begin{array}{c} {\tilde{u}}_{1}\\ {\tilde{u}}_{2} \end{array}\right]=N_{m}\left[\begin{array}{c} u_{1k}\\ u_{2k} \end{array}\right],$$
where ${\tilde{u}}_{1}$, ${\tilde{u}}_{2}$ are the element interpolated displacements, $N_{m}$ is the shape functions matrix for a 4-nodes membrane element and subscript k∈{1,2,3,4} refers to the specific node. The bending DOFs, representing the out-of-plane displacements and rotations, are interpolated by applying bending shape functions, as shown in Equation (Equation 4).
(4)$$\left[\begin{array}{c} {\tilde{u}}_{3}\\ {\tilde{\phi }}_{1}\\ {\tilde{\phi }}_{2} \end{array}\right]=N_{b}\left[\begin{array}{c} u_{3k}\\ \phi_{1k}\\ \phi_{2k} \end{array}\right],$$
where $N_{b}$ is the shape functions matrix related to bending DOFs.

The stiffness matrix in local element coordinates is obtained by assembling the membrane matrix (Equation (Equation 5)), corresponding to the two in-plane translational displacements ($u_{1}$ and $u_{2}$), and the bending matrix (Equation (Equation 6)), related to the flexural behaviour of the thin-plane element, which consists of 3 DOFs ($u_{3}$, $\phi_{1}$ and $\phi_{2}$). The sixth DOF, $\phi_{3}$, related to the twisting response, is assigned an arbitrary stiffness much lower than the rest of the components, taking into consideration that this rotation does not contribute to strain energy [22]. Nevertheless, this DOF is required for the consistency of matrices when transforming to the global coordinate system. The integration in the domain of the element ($\Omega^{e}$) is reduced to an integration in the area (*A*), described in the $x_{1}$ and $x_{2}$ directions. This integration is performed numerically by applying a reduced integration based on the Gaussian quadrature rule to avoid shear locking.
(5)$$K_{m}=\int_{\Omega^{e}}B_{m}^{T}E_{m}B_{m}\;d\Omega^{e}=h\int_{A}B_{m}^{T}E_{m}B_{m}\;dA$$
(6)$$K_{b}=\frac{h^{3}}{12}\int_{A}B_{b}^{T}E_{b}B_{b}\;dA+hK_{t}\int_{A}B_{s}^{T}E_{s}B_{s}\;dA,$$
where B is the derivative of the shape functions, E the material stiffness tensor, particularized in this study for a linear isotropic elastic material, and *h* is the thickness of the element (dimension in $x_{3}$-direction). The subscripts *b*, *m* and *s* represent bending, membrane and shear, respectively. Finally, $K_{t}$ represents the stiffness associated with the drilling DOF ($\phi_{3}$), which value is about one-thousandth of the smallest diagonal element of the element matrix stiffness, following recommendations in literature [22].

As can be seen in Equation (Equation 1), the computation of the electric charge requires the calculation of the integral of the in-plane strains of the element. If this calculation is defined in element local coordinates, the Equation (Equation 7) is obtained.
(7)$$\left[\begin{array}{c} {\tilde{\varepsilon }}_{11}\\ {\tilde{\varepsilon }}_{22}\\ {\tilde{\varepsilon }}_{12} \end{array}\right]=\left[\begin{array}{c} \frac{\partial {\tilde{u}}_{1}}{\partial x_{1}}\;\\ \frac{\partial {\tilde{u}}_{2}}{\partial x_{2}}\;\\ \frac{\partial {\tilde{u}}_{1}}{\partial x_{2}}+\frac{\partial {\tilde{u}}_{2}}{\partial x_{1}} \end{array}\right]-x_{3}\left[\begin{array}{c} -\frac{\partial^{2}{\tilde{u}}_{3}}{\partial x_{1}^{2}}\;\\ \frac{\partial^{2}{\tilde{u}}_{3}}{\partial x_{2}^{2}}\;\\ 2\frac{\partial^{2}{\tilde{u}}_{3}}{\partial x_{1}\partial x_{2}} \end{array}\right]=B_{m}\left[\begin{array}{c} u_{1k}\\ u_{2k} \end{array}\right]-x_{3}B_{b}\left[\begin{array}{c} u_{3k}\\ \phi_{3k}\\ \phi_{3k} \end{array}\right].$$

Newton’s second law for a system leads us to:(8)$$K\bar{u}+C\dot{\bar{u}}+M\ddot{\bar{u}}=0,$$
where C is the damping matrix (C=0 in our problem). Assuming that the structure undergoes harmonic vibration, the modal coordinates are defined by means of $\eta_{*i}=\sin\left(\omega_{i}t\right)$, where $\omega_{i}$ is the natural frequency corresponding to the *i*-th vibration mode.
(9)$$\left(K-\omega_{i}^{2}\;M\right)\;\varphi_{i}=0.$$

The mass matrix M is computed in the same way as the stiffness matrix K, assembling the matrices corresponding to the membrane and bending, presented in Equations (Equation 10) and (Equation 11).
(10)$$M_{m}=\int_{\Omega^{e}}\rho N_{m}^{T}N_{m}\;d\Omega^{e}=h\rho \int_{A}N_{m}^{T}N_{m}\;dA$$
(11)$$M_{b}=\int_{A}N_{b}^{T}PN_{m}\;dA,$$
where ρ is the material density and $P=\rho \left[\begin{array}{ccc} h & 0 & 0\\ 0 & \frac{h^{3}}{12} & 0\\ 0 & 0 & \frac{h^{3}}{12} \end{array}\right]$.

## 3. Formulation of the Problem

### 3.1. Electrode Profile

The initial formulation of the problem involves the use of the tri-level function χ∈{−1,0,1} to represent negative, null, or positive polarity. Once the problem has been discretized in finite elements, this function can take any of these three values, but remains constant on the element.

The usual topology optimization problem uses a relaxation method over the design variable. This allows the variable to take any value between the minimum and the maximum allowed for each element *e*:$$\chi_{e}\in \left\{-1,0,1\right\}\to \varphi_{e}\in \left[-1,1\right].$$

The relaxation of the design variable often results in the well-known problem of grey areas. In mechanical problems where the design variable represents structure/void, this is understood as microstructure, that is, a mixture of solid and void. The most common procedure to alleviate this difficulty is the use of a penalization (also called power) to enforce the variable to take extreme values.

For this particular problem, the behaviour of the design variable φ is quite different. As can be seen in the next section, the linear dependence of the objective function with respect to the optimization variable leads us to “black and white” designs, where the microstructure commented above does not appear.

### 3.2. Optimization Problem

Having in mind the maximization of the electric charge, the discretized expression for computing this parameter is presented:(12)$$q=J^{T}\left(\varphi \right)U=\sum_{e}^{n_{el}}\varphi_{e}B_{e}^{T}U_{e},$$
with $n_{el}$ the number of finite elements, $B_{e}$ the discretization of the strain displacement matrix, $U_{e}$ is the displacements vector (solution of the equilibrium equation) of the element *e*. It is important to emphasize that the relaxed variable $\varphi_{e}$ does not need any kind of regularization (filtering techniques). The piezoelectric property $e_{31}$ has been removed from the objective function since a constant does not affect the optimal design. The vector U, corresponding to the displacement of the structure, is computed according to the nature of the problem. Finally, the optimization problem states as follows:$$\max_{\varphi }:\;\;\;\;q,$$
subject to:$$\left\{\begin{array}{c} \begin{array}{cccc} KU & = & F & \mathrm{static}\;\;\mathrm{case}\;\\ \left(K-\omega_{i}^{2}M\right)U_{i} & = & 0 & \mathrm{vibration}\;\;\mathrm{modes}\;\\ \varphi & \in & \left[-1,1\right] \end{array} \end{array}\right.,$$
where subscript *i* represents the *i*-th vibration mode, with $\omega_{i}$ the natural frequency and $U_{i}$ the modal shape. φ is the relaxed variable, that can take values between [−1,1] to define the polarity of the electrode profile.

In practice, the polarization variable φ does not need any kind of regularization, in the sense that it takes the extreme values without using filtering techniques [16]. The objective function depends linearly on the design variable and then, the calculus of the derivatives has not been included.

## 4. Numerical Examples

The domain design proposed is a conical-shaped shell, due to its applications in the aerospace industry. The geometry and material are based on the study by Sun et al. [11]. The radii of the top and bottom sides are $r_{t}=100$ mm and $r_{b}=150$ mm, respectively. The height is $h_{c}=500$ mm and the wall thickness $t_{c}=1$ mm. The material used in the cone is aluminium with Young’s modulus $E_{Al}=70$ GPa, Poisson’s ratio $\nu_{Al}=0.33$ and mass density $\rho_{Al}=2600\;\mathrm{Kg}/m^{3}$. Due to the linear behaviour of the problem, the value of the piezoelectric constant is fixed to $e_{31}=1\;C/m^{2}$.

The displacement field is computed in both case studies using the commercial software COMSOL Multiphysics 5.4 [24] and checked with Abaqus 2019 [25].

### 4.1. Static Case

Two different distributions of loads are studied in this section. Both correspond to different situations that can be found when this kind of sensors is used in the aerospace industry. In general, the real load state of a structure is a combination of the different cases presented in this paper, but our objective is to analyse the results separately.

#### 4.1.1. Pressure Uniformly Distributed

The objective of this section is to analyse the behaviour of the structure when it is subjected to a constant pressure load over its entire surface, simulating pressurization. The displacement at the meridional direction of the cone is fixed on the top edge. The pressure applied is $p_{0}=1\cdot 10^{5}$ Pa (positive referring to external compressive pressure). The deformed structure, the mapping of the electric charge and the optimized electrode profile are depicted in Figure 1.

The displacement is scaled to be able to see the deformation of the structure. The value of $\varepsilon_{11}+\varepsilon_{22}$ (the electric charge depends linearly on it) in each point of the surface is shown in Figure 1 (centre). The electrode profile is homogeneous, φ=−1, meaning that the whole structure is covered with a unique electrode with the same polarity (blue colour in Figure 1 (right)). The total electric charge generated by the sensor is $q=3.8254\cdot 10^{-5}$ C. Thanks to the assumption of linear elasticity, and the linear dependence of the objective function with the variable φ, a change in the sign of the pressure $p_{0}$, leads us to a change in the sign of the electrode profile φ=1. In such a case, the electric charge remains constant.

#### 4.1.2. Punctual Forces

In this loading scenario, the boundary of the upper side of the cone is clamped, i.e., the rotations and the displacements are fixed, while the rest of the structure is free. The punctual forces are applied at four points equally distributed around the bottom edge. For the first load case, the forces are pointing away from the cone, with a value of $f_{in}=100$ N. The deformed shell and the optimized electrode are shown in Figure 2.

The value of the electric charge generated by the sensor is $q=5.1267\cdot 10^{-5}$ C. The displacement shown in Figure 2 is scaled by a factor of 20, otherwise, the deformation would be too small to appreciate the deformed shape.

The violet colour in the deformation (Figure 2 (left)) represents the biggest displacement (in magnitude), while the white part shows the smallest one, near the boundary conditions. The electrode profile (Figure 2 (right)) does not map the displacement field, but the curvature of the structure, in the sense that the value of the variable φ is directly related with the sign of $\varepsilon_{11}+\varepsilon_{22}$. Blue and white represent electrodes with opposite polarization. The sign of the variable is computed element by element, and then interpolated to the whole surface. This means that in very small areas, φ can take null values where $\varepsilon_{11}+\varepsilon_{22}=0$ (orange colour in Figure 2 (right)). These could be taken as small gaps that are also needed to avoid short circuits.

A second load case is presented in Figure 3. The two forces applied over the *y*-axis change their sense of application.

As happened in the previous case, small parts of the surface exhibit φ=0, where an alternation of the polarization of the electrode occurs. The value of the objective function, in this case, is $q=3.7736\cdot 10^{-5}$ C.

### 4.2. Vibration Modes

The electrode profile for the first four vibration modes (with increasing natural frequencies) is obtained in this section. Once again, the cone is clamped at its upper side. The mode shapes (eigenvectors of the equilibrium equation) are *M*-orthonormalized:$$U_{i}^{T}MU_{j}=\delta_{ij}.$$

This means that the amplitude of the vibration is fixed by this normalization, and then, the electric charge produced by different vibration modes cannot be compared. It is convenient to establish a different normalization to check that the optimized electrode profile is improving the response of the sensor. With this objective, a first attempt at normalizing the objective function is presented:$$q_{norm,i}=\frac{e_{31}\sum_{e}^{n_{el}}\varphi_{e}B_{e}^{T}U_{e,i}}{\left|\sum_{e}^{n_{el}}B_{e}^{T}U_{e,i}\right|}.$$

The parameter $q_{norm,i}$ directly give information about the improvement of the electric charge produced when the electrode profile is optimized for the *i*-th vibration mode. The main issue using this normalization is the symmetry of the vibration mode, in the way that parts with different polarities are compensated, and then, the electric charge generated is close to zero. Therefore, the normalization with respect to the maximum displacement of the mode shape is proposed:$${\tilde{q}}_{norm,i}=\frac{e_{31}\sum_{e}^{n_{el}}\varphi_{e}B_{e}^{T}U_{e,i}}{\max\left(|U_{i}|\right)}$$

The parameter ${\tilde{q}}_{norm,i}$ gives us information about the charge produced per unit of displacement, but it is important to remark that this is not comparable between the two modes.

For the first vibration mode, $\left(\omega_{1},U_{1}\right)$, the mode shape, the mapping of the electric charge and the electrode profile are shown in Figure 4.

The value of objective function is ${\tilde{q}}_{norm,1}=1.6961$ C/m. The value of the charge depicted in the centre figure is normalized, meaning that red and blue colours represent an electric charge of 1 and −1, respectively.

The same process is repeated for the second (Figure 5), the third (Figure 6) and the fourth (Figure 7) vibration modes.

Firstly, note that the modal shapes and frequencies correspond with the work of Sun et al. [11], which serves to verify the results obtained with our numerical model. Secondly, the modal charges produced for each vibration mode are: ${\tilde{q}}_{norm,2}=1.8219$ C/m, ${\tilde{q}}_{norm,3}=4.6121$ C/m and ${\tilde{q}}_{norm,4}=1.2088$ C/m. Although these values have no real physical meaning as they depend on the eigenvector, they may be used in a modal decomposition to compute the participation of each vibration mode.

## 5. Conclusions

This work presents the relationship between the deformation obtained in a cone-shaped sensor and the electric charge collected. The electrode profile placed over the piezoelectric layer is optimized to maximize the electric response of the device.

Considering real applications in the aerospace industry, two different study cases are analysed: static loading and vibration modes. The former is divided into two situations, pressure uniformly distributed and punctual loads applied at the bottom edge. In general, the sensor will be subjected to both kinds of forces, although they are studied independently. The latter analyses are based on the free vibration of the cone, to compute the participation of each mode for a pure dynamic case.

Although a fully realistic loading situation has not been included in the paper, the method proposed can optimize the electrode profile when static and dynamic forces are applied. This combination of cases will produce a deformation that is the control of the optimization problem.

In both cases, the electric behaviour of the sensor is optimized through a tri-level function that represents the polarity of the electrode profile. It is demonstrated from the analytical expressions, that the sign of this variable (electrode) must coincide with the sign of the sum of in-plane strains, to maximize the objective function.

The sensor is modelled under the assumption of the linear elasticity theory for thin conical shells, but it could be extended to other geometries, large displacements (geometrically non-linear modelling) and/or large strains.

## Figures and Tables

**Figure 1 sensors-23-00442-f001:**
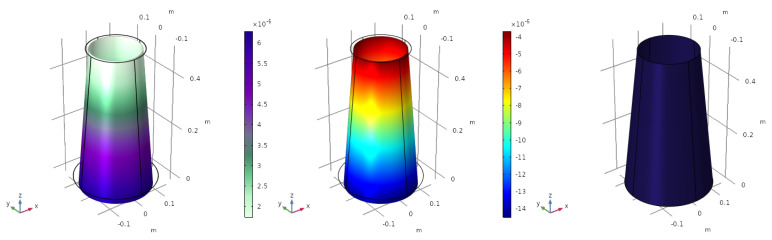
Pressure uniformly distributed. Deformation of the structure (**left**), electric charge (**centre**) and optimized electrode profile (**right**).

**Figure 2 sensors-23-00442-f002:**
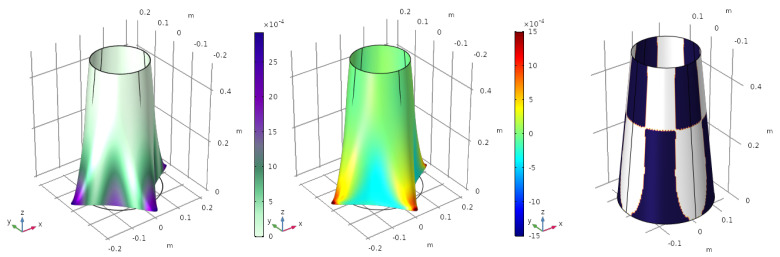
First load case. Deformation of the structure (**left**), electric charge (**centre**) and optimized electrode profile (**right**).

**Figure 3 sensors-23-00442-f003:**
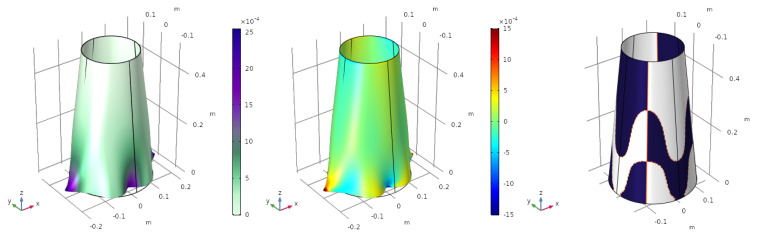
Second load case. Deformation of the structure (**left**), electric charge (**centre**) and optimized electrode profile (**right**).

**Figure 4 sensors-23-00442-f004:**
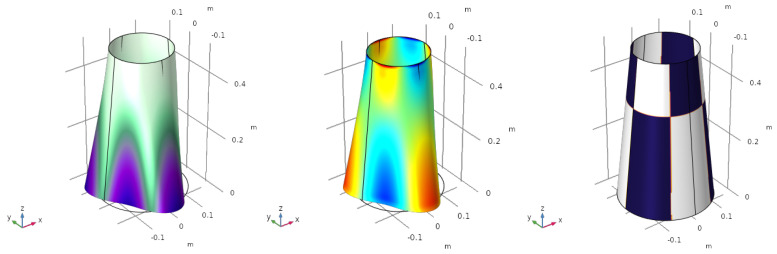
First vibration mode $f_{1}=147$ Hz. Mode shape (**left**), electric charge (**centre**) and optimized electrode profile (**right**).

**Figure 5 sensors-23-00442-f005:**
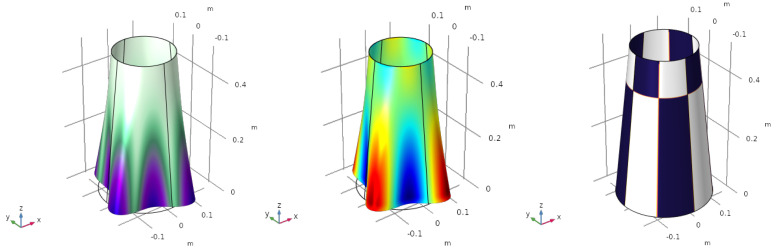
Second vibration mode $f_{2}=198$ Hz. Mode shape (**left**), electric charge (**centre**) and optimized electrode profile (**right**).

**Figure 6 sensors-23-00442-f006:**
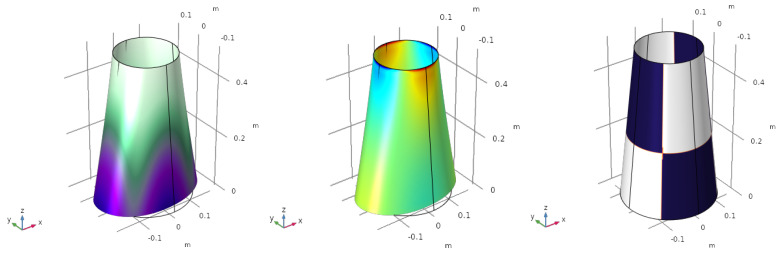
Third vibration mode $f_{3}=222$ Hz. Mode shape (**left**), electric charge (**centre**) and optimized electrode profile (**right**).

**Figure 7 sensors-23-00442-f007:**
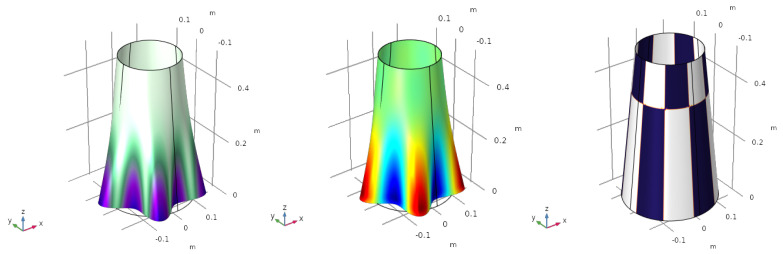
Forth vibration mode $f_{4}=299$ Hz. Mode shape (**left**), electric charge (**centre**) and optimized electrode profile (**right**).

## Data Availability

Not applicable.

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
