# Peer review of "Optimization of the Polarization Profile of Conical-Shaped Shells Piezoelectric Sensors"

_sensors, 2022, doi:10.3390/s23010442_

Round 1

Reviewer 1 Report

1.Many researches have been done on the surface load distribution monitoring of some parts by using piezoelectric sensors. The innovation point of this article is what, had better have relevant contrast.

2. In this paper, the optimal arrangement of piezoelectric elements under 7 different conditions is proposed, but whether the piezoelectric elements can be arranged in this way in practice, for example, fig. 2 and Fig. 3, whether the irregular electrode arrangement is difficult to design. At the same time, how to arrange the lead wire of multi-sensor, hope to combine the actual situation to explain.

3. In this paper, 7 different loading conditions of static and free vibration are studied, and an arrangement scheme is proposed for each loading condition, but in practice, there may be many loading conditions, such as vibration mode, the deformation under four different frequencies is analyzed, but there may be different frequencies in practice, so whether the study of this paper is of practical significance and whether the seven permutations should be considered synthetically.

4.Under static loading, what is the difference between fig.2 and fig.3, and how the structure is deformed and the optimized electrode profile is transformed, whether the actual electrode can be arranged like this, as fig.3.

5. Under the vibration mode, the deformation of the structure under four different frequencies is analyzed, why choose 147 Hz, 198 Hz, 222 Hz, 299 Hz four frequencies to study, what basis.

6. The graph data given in this paper is less, so the charge quantity of each arrangement should be calculated, and the corresponding curve should be drawn, which is more intuitionistic.

Reviewer 2 Report

The work reports the methodology and topological optimization of the conical-shaped shells piezoelectric sensors. I believe that this manuscript is well organized and written. I recommend it to be published.

Reviewer 3 Report

I have attached the file in this email.

Round 2

Reviewer 1 Report

I think I can accept your revised paper for publication.

Reviewer 3 Report

The authors have correctly made all the required changes.